# Three-dimensional covalent organic frameworks with pto and mhq-z topologies based on Tri- and tetratopic linkers

Dongyang Zhu [1,8], Yifan Zhu [2,8], Yu Chen[1], Qianqian Yan [2], Han Wu[3], Chun-Yen Liu [1], Xu Wang[4], Lawrence B. Alemany[4,5], Guanhui Gao[2,4], Thomas P. Senftle [1], Yongwu Peng [6], Xiaowei Wu [7] ✉ & Rafael Verduzco [1,2] ✉

Three-dimensional (3D) covalent organic frameworks (COFs) possess higher surface areas, more abundant pore channels, and lower density compared to their two-dimensional counterparts which makes the development of 3D COFs interesting from a fundamental and practical point of view. However, the construction of highly crystalline 3D COF remains challenging. At the same time, the choice of topologies in 3D COFs is limited by the crystallization problem, the lack of availability of suitable building blocks with appropriate reactivity and symmetries, and the difficulties in crystalline structure determination. Herein, we report two highly crystalline 3D COFs with pto and mhq-z topologies designed by rationally selecting rectangular-planar and trigonal-planar building blocks with appropriate conformational strains. The pto 3D COFs show a large pore size of 46 Å with an extremely low calculated density. The mhq-z net topology is solely constructed from totally face-enclosed organic polyhedra displaying a precise uniform micropore size of 1.0 nm. The 3D COFs show a high $CO_2$ adsorption capacity at room temperature and can potentially serve as promising carbon capture adsorbents. This work expands the choice of accessible 3D COF topologies, enriching the structural versatility of COFs.

Covalent organic frameworks (COFs) are a class of porous and crystalline polymers that are assembled molecularly from organic building blocks[1–3]. COFs have attracted significant attention for versatile applications such as gas storage, water remediation, catalysis, energy storage, and drug delivery due to their tunable chemistry, high porosity, high surface areas, and good stability[4–9]. Precise construction of organic building units can produce COFs with two-dimensional (2D) and three-dimensional (3D) topologies[10]. While a rich library of planar building units and straightforward crystallization processes have been developed for 2D COFs[11], examples of 3D COFs are much more limited[12–14]. This is due to the limited number of building blocks that have been discovered for producing 3D COFs, the limited set of

[1]Department of Chemical and Biomolecular Engineering, Rice University, 6100 Main Street, MS-362, Houston, TX 77005, USA. [2]Department of Materials Science and NanoEngineering, Rice University, 6100 Main Street, MS-325, Houston, TX 77005, USA. [3]Ganjiang Chinese Medicine Innovation Center, Nanchang 330000, China. [4]Shared Equipment Authority, Rice University, 6100 Main Street, Houston, TX 77005, USA. [5]Department of Chemistry, Rice University, 6100 Main Street, Houston, TX 77005, USA. [6]College of Materials Science and Engineering, Zhejiang University of Technology, Hangzhou 310014 Zhejiang, China. [7]Xiamen Key Laboratory of Rare Earth Photoelectric Functional Materials, Xiamen Institute of Rare Earth Materials, Fujian Institute of Research on the Structure of Matter, Haixi Institutes, Chinese Academy of Sciences, Xiamen 361021, China. [8]These authors contributed equally: Dongyang Zhu, Yifan Zhu. ✉e-mail: xmwuxiaowei@fjirsm.ac.cn; rafaelv@rice.edu

topologies, and synthetic challenges in producing crystalline 3D organic frameworks[12]. 3D COFs are typically constructed from highly connected 3D geometry building units with prefixed spatial orientation, such as (4/6/8-connected) building nodes with tetrahedral/triangular prism/cubic symmetries[15–19]. However, synthesis of polyhedral organic building blocks with high valency (>5) can be challenging and laborious since this strategy heavily relies on the organic chemistry of sp$^2$ and sp$^3$ hybridization, which typically only produces the valency of 3 and 4[20]. Another strategy for constructing 3D COFs is to precisely control the conformational flexibility of the linkages to realize specific topologies[21–23]. These strategies have been implemented to construct 3D COFs with nbo[21], srs[22] and rra[23] topologies, in which tetrahedral SiO$_4$ or BO$_4$ was used as the predesigned rigid linkages to produce the desired nets. However, construction of 3D COFs is still challenging due to the limited number of flexible linkages. Therefore, there is an urgent need to further develop 3D COFs and extend the building block library.

Reticular chemistry enables the linking of molecular building blocks into extended crystalline structures through strong bonds[24] and it has been used to build 3D COFs from planar molecules. For example, the combination of square-planar (4-c) and trigonal-planar (3-c) building units can produce multiple high-symmetry topologies including 2D nets like bex[25], tth[26], mtf[27], or 3D pto, tbo, mhq-z, fjh, iab, gee, and ffc topologies (Fig. 1) due to the flexible conformation or dihedral angle imposed on their linkages[28]. Some of these topologies have been previously elucidated. For example, Zhong and co-authors reported the discovery of 3D COFs with an ffc topology by connecting tetraamines with trialdehydes[29]. Cui and coworkers reported 3D COFs with tbo topology through the assembly of square-planar porphyrin amine and trigonal-planar aldehyde[30]. A series of fjh topology 3D COFs with isoreticular structures were also reported by Yaghi and coworkers through the rational conformational design of building blocks[28].

However, other topologies composed of planar [4 + 3] combinations like pto, mhq-z, iab, and gee nets have not been observed for 3D COFs.

From the crystallographic point of view, when the highest symmetry is preferred, the combination of planar 4-c and 3-c linkers will generate two types of vertices and one type of edge[31,32], leading to three 3D edge-transitive nets as tbo, pto and mhq-z[33]. Herein, we report the discovery of 3D COFs with pto and mhq-z topologies. We judiciously selected rectangular-planar (4-c) and trigonal-planar (3-c) building blocks with appropriate conformational strains when they assemble into 3D networks (Figs. 1 and 2). We systematically analyzed the crystal structures of these COFs by comparing experimental PXRD with all possible simulated structures. The pore structures of refined models were consistent with the pore size distributions derived from nitrogen sorption tests. The pto 3D COFs show a large pore size (46 Å), low density of 0.0943 g cm$^{-3}$, and very high porosity (94.2%) when compared with other crystalline framework materials. The mhq-z net COF is solely constructed from face-enclosed organic polyhedra, producing a uniform micropore size of 1.0 nm. Some of the as-prepared COFs demonstrated high CO$_2$ adsorption capacity at room temperature and may potentially serve as promising carbon capture adsorbents. This work represents a successful construction of 3D COFs through the rational design of planar building blocks. The unique pore shapes of the reported 3D COFs my lead to promising gas adsorption performance and further promote their applications as efficient adsorbents.

## Results

As shown in Fig. 2, RICE-3 was synthesized through polycondensation of trigonal-planar 1,3,5-tris(4-aminophenyl)benzene (TAPB) and rectangular-planar 4′,4‴,4‴‴,4‴‴‴-(ethene-1,1,2,2-tetrayl)tetrakis(([1,1′-biphenyl]-4-carbaldehyde)) (ETTBC) in a solvent mixture of dioxane/mesitylene/6 M acetic acid (AcOH) (4/1/1, v/v/v) at 120 °C for 3 days.

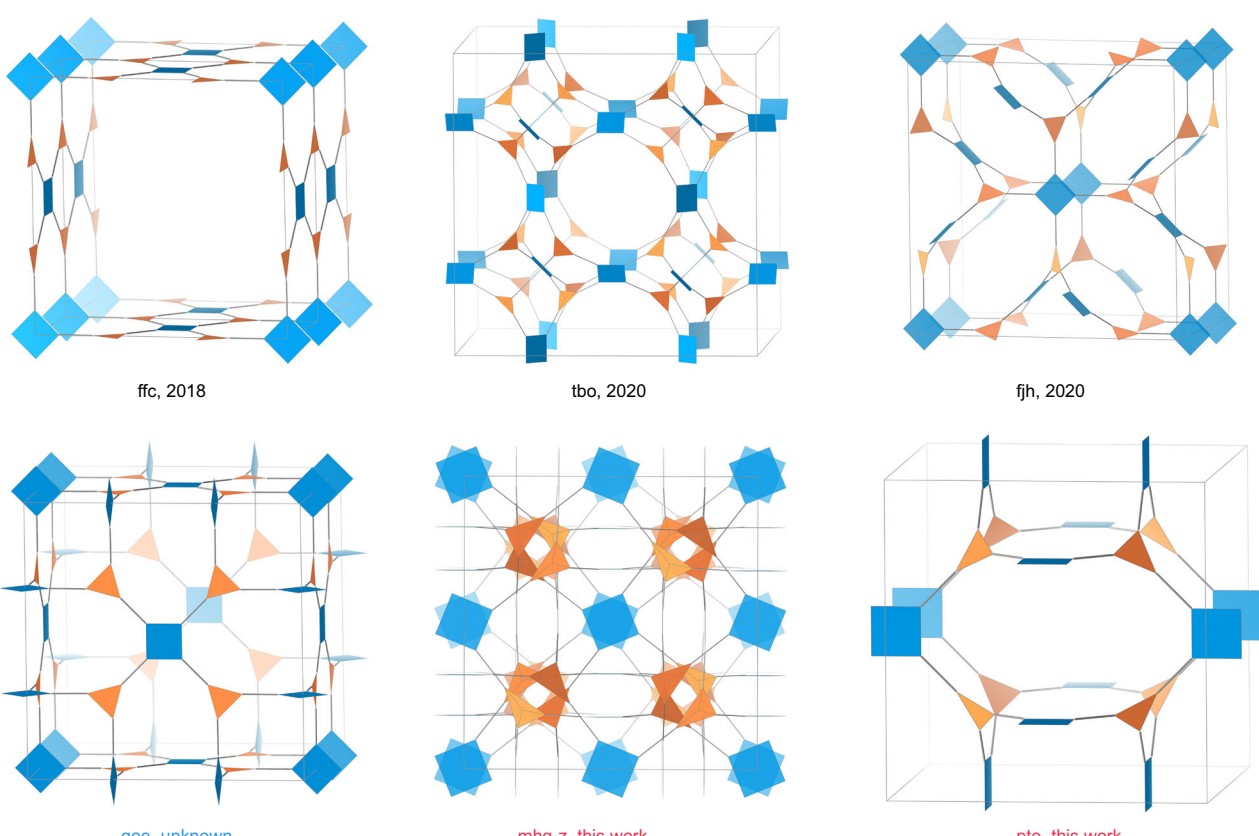

ffc, 2018          tbo, 2020          fjh, 2020

gee, unknown          mhq-z, this work          pto, this work

**Fig. 1 | 3D COFs constructed from 4-c and 3-c building blocks.** 3D COF topologies constructed from 4-c and 3-c building blocks. This manuscript reports 3D COFs with mhq-z and pto topologies.

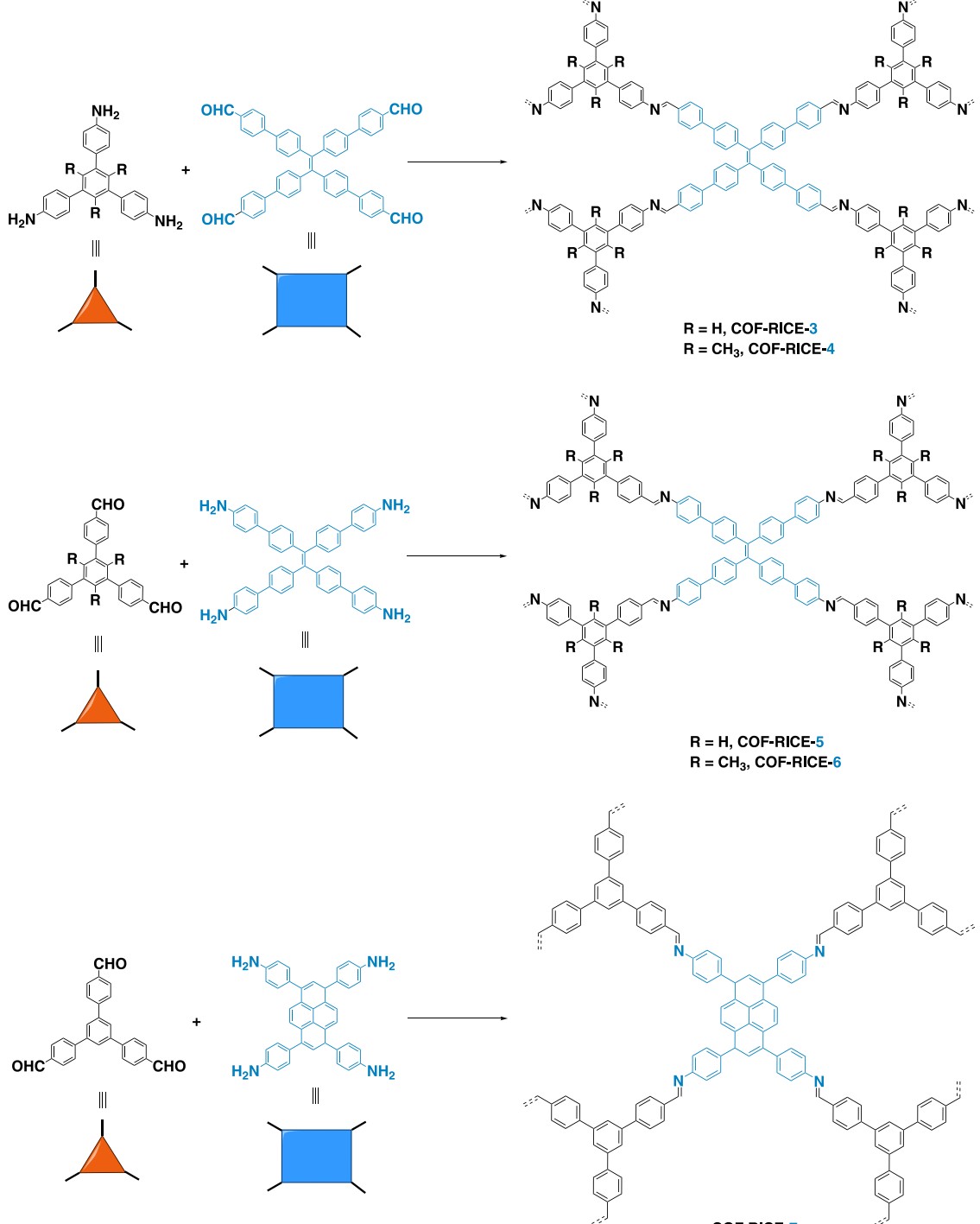

**Fig. 2 | Synthesis of RICE-3/4/5/6/7.** Synthesis scheme for the preparation of 3D imine-linked COFs. All COFs prepared under solvothermal reaction conditions.

The precipitates were thoroughly washed using tetrahydrofuran (THF), acetone, and ethanol followed by supercritical $CO_2$ drying to produce RICE-3 as yellowish crystalline powders with an approximately 72% yield. Isoreticular structures of RICE-4/5/6 were synthesized using a similar procedure, but the reaction was conducted in a solvent mixture of o-dichlorobenzene (o-DCB)/1-butanol (1-BuOH)/6 M AcOH (5/5/2) at 120 °C for 7 days. RICE-7 was synthesized by condensation of 1,3,6,8-tetrakis(4-aminophenyl)pyrene (Py) and 1,3,5-tris(4-formylphenyl) benzene (TFPB) in a solvent mixture of o-DCB/1-BuOH/6 M AcOH (5/5/1) at 120 °C for 7 days. RICE-7 was separated as dark yellow powders with an isolated yield of 95%. The detailed synthesis and activation

procedures and screening of reaction conditions are presented in Supplementary Information Figs. 1–5. All five COFs were insoluble in common organic solvents such as dimethyl sulfoxide (DMSO), dimethylformamide (DMF), THF, acetone, methanol, and hexane.

The successful polycondensation to form imine linkages was demonstrated by the combination of Fourier-transform infrared spectroscopy (FTIR) (Supplementary Figs. 6–10) and solid-state $^{13}C$ cross-polarization/magic-angle spinning (CPMAS) nuclear magnetic resonance (NMR) spectroscopy (Supplementary Figs. 11–15). As displayed in Supplementary Fig. 6, the N-H stretching vibration peaks at 3440 cm$^{-1}$ and 3358 cm$^{-1}$ and the C=O stretching vibration peak at 1692

cm$^{-1}$ disappeared in the final COF materials, while the C=N stretching vibration peak at 1620 cm$^{-1}$ appeared, indicating imine bond formation[34–36]. Solid-state $^{13}$C CPMAS NMR spectroscopy of RICE-3 exhibited prominent signals located at 158.5 ppm (Supplementary Fig. 11), indicating the presence of carbons from imine groups. Meanwhile, only weak aldehyde carbonyl signals could be observed, which could be attributed to unreacted aldehyde groups on the edges of the COFs[34,35]. Similar imine carbons signals could also be observed in solid-state $^{13}$C CPMAS NMR spectroscopy of RICE-4 to RICE-7 (Supplementary Figs. 12–15). Scanning electron microscopy (SEM) images (Supplementary Fig. 19) showed that RICE-3/5/6/7 were comprised of uniform aggregated microspheres, while RICE-4 was composed of stacking microrods. Thermogravimetric analysis (TGA) indicated that the COFs possessed high thermal stability and did not degrade until approximately 450 °C (Supplementary Fig. 16).

The precise structures of COFs were elucidated through a combination of PXRD analyses and crystallographic architectures (Fig. 3a and Supplementary Figs. 32–54) with the aid of pore size distributions derived from nitrogen sorption tests (Fig. 3c and Supplementary Figs. 59–69). RICE-3 exhibited a PXRD pattern (Fig. 3a) with several sharp peaks and low background, where the main peaks at 1.76°, 3.07°, 4.02°, and 5.37° can be precisely indexed as (100), (111), (210) and (300) facets. Based on the basic concept of reticular chemistry, the combination of tetratopic-planar (4-c) and trigonal-planar (3-c) monomers can be fully reticulated as tbo, pto, ffc, fjh, gee, mhq-z, etc[37]. or be partially covalently linked as a sub-stoichiometric 2D net such as bex[25]. We built those alternative models accordingly (Supplementary Figs. 37–46) and compared them with the experimental one (Supplementary Fig. 37), and the results suggested that RICE-3 adopted the unprecedented pto net with cubic Pm-3 space group (Fig. 3b). Rietveld refinement was conducted to obtain the unit cell parameters $a = b = c = 49.42$ Å, $\alpha = \beta = \gamma = 90°$). The experimental PXRD pattern shows good agreement with the simulated one, yielding a good agreement and fair profile differences. RICE-4/5/6 showed similar XRD patterns as RICE-3, indicating their isoreticular pto net (Supplementary Figs. 47–49).

The validity of the proposed isoreticular pto net crystalline structures was further confirmed by $^1$H NMR spectra of digested COF samples (Supplementary Figs. 75 and 76, see details in Supplementary Information). RICE-3 was fully digested in a solvent mixture of DMSO-$d_6$ and deuterium chloride solution (35 wt % in D$_2$O, >99 atom % D) (v/v, 10/1), so we could calculate the linker ratio from the solution NMR measurements, which should be equal to the monomer ratio in the COF structure. The characteristic peaks of both aldehyde and amine monomers were observed in the corresponding regions on $^1$H NMR, and the ratio of ETTBC to TAPB was calculated to be approximately 3:4 (Supplementary Fig. 75), which was consistent with the structural model and excluded the possible sub-stoichiometric 2D net structure. Elemental analysis of RICE-3 (Calcd: C, 90.18; H, 4.93; N, 4.89%. Found: C, 90.0; H, 4.1; N, 4.4%) was also in good agreement with expected monomer ratios of 3:4 (aldehyde to amine), which further confirms the 3-4 connected framework model.

We further tested the N$_2$ sorption behavior of these COFs (RICE-3-6) (Fig. 3, Supplementary Figs. 64–69 and 72–74). All the COFs displayed high N$_2$ uptake, and the N$_2$ adsorption capacity increased rapidly at the lower relative pressure range ($P/P_O < 0.1$), characteristic of microporous materials. Hysteresis can be observed at $P/P_O > 0.4$), indicative of mesoporous structures. The Brunauer−Emmett−Teller (BET) surface area of RICE-3 was calculated to be 720 m$^2$ g$^{-1}$, and pore size distributions estimated using quenched solid density functional theory (QSDFT) model (slit/cylindrical pore at the adsorption branch) gave values centered around 1.0 nm, 1.4 nm, 3.2 nm, and 4.6 nm (Fig. 3c), consistent with our simulated structure (Fig. 3d). Impressively, due to the highly porous structures of pto frameworks, RICE-3 demonstrated a large mesoporous pore aperture of approximately 4.6 nm (Fig. 3e), larger than that of other reported 3D COFs (see

Supplementary Tab. 1)[38–41]. Note that pto is an open porous edge-transitive and highly symmetric 3D structure, which shows the same apertural shapes when viewed from three orthogonal vectors (Fig. 3f). The calculated densities (0.0943 to 0.0995 g cm$^{-3}$) and porosities (92.0% to 94.2%) for RICE-3/4/5/6 are lower and higher, respectively, than other crystalline framework materials, including MOFs[32] and COFs (Supplementary Tab. 2).

PXRD patterns of RICE-7 exhibited several discernable peaks (facets) at 3.84° (111), 4.28° (200), 5.09° (220), 6.44° (222), 8.98° (422), and 11.10° (442), and we were able to index at least ten peaks from the PXRD pattern (Fig. 4a). We performed structural modeling based on the potential [4 + 3] combination nets, such as pto, tbo, ffc, and mhq-z (Supplementary Figs. 51–59), and determined that the most credible structure for RICE-7 was an mhq-z topology with cubic F23 space group (Fig. 4b). Rietveld refinement against the experimental PXRD pattern was carried out to calculate the unit cell parameters ($a = b = c = 46.01$ Å, $\alpha = \beta = \gamma = 90°$). The ratio of TFPB to Py was also calculated to be 4:3 from the digested RICE-7 (see Supplementary Fig. 77 and other details in Supplementary Information), which was in line with the stoichiometric ratio found in the mhq-z net. This ratio was also very close to that calculated from the elemental analysis of RICE-7 (Calcd: C, 89.91; H, 4.57; N, 5.52%. Found: C, 86.7; H, 3.3; N, 5.2%). Furthermore, the high-resolution transmission electron microscopy (HRTEM) images (Supplementary Figs. 33 and 35) showcased several crystallites with periodic channel-like features (Supplementary Figs. 29 and 31), and the channel widths of 2.0 nm and 0.5 nm agreed favorably with the 1D channel distance in crystal structures (details in Supplementary Figs. 32–36).

It is worth noting that the mhq-z topology has not been observed in COFs and is rare in other reticular frameworks, including metal organic frameworks (MOFs) and zeolites[42,43]. The BET surface area of RICE-7 was estimated to be 1448 m$^2$ g$^{-1}$, and the application of the QSDFT model to estimate pore size from N$_2$ adsorption isotherm affords narrow pore width distribution that centered at 1.0 nm (Fig. 4c). Truncated polyhedra cages in the COF structures stacked together and formed 1D open channels with 1.0 nm diameter (Fig. 4d). Visualizations of these pore channels for RICE-7 from different facets are provided in Supplementary Figs. 60, 61.

RICE-7 is composed of three types of truncated polyhedra cages by sharing faces. Specifically, four TFPB monomers are closely assembled to form a small truncated tetrahedron cage (T$^{cage}$) with a diameter of 11 Å (Fig. 5a). The largest cage is a closed truncated cube (C$^{cage}$), which is constructed from six Py and eight TFPB monomers to form a large cage with a diameter of 29 Å (Fig. 5c), and eight T$^{cages}$ are located at eight corners of the pristine C$^{cage}$ by sharing each TFPB window. Another cubic B$^{cage}$ (diameter of 18 Å) is also formed from six Py monomers (Fig. 5b), which serves as a bridge to connect the neighbor C$^{cage}$ (Fig. 5d) and results in the mhq-z topology (Fig. 5f). Interestingly, all three polyhedra cages are completely enclosed (Fig. 5e) without any open window to form 1D channels, which has not been discovered in COFs materials. Only one channel of 1.0 nm size is observed from the packing of those cages (Fig. 4d), which is consistent with the N$_2$ sorption result.

The CO$_2$ adsorption behaviors of the 3D COFs were further studied to expand their potential applications in greenhouse gas capture (Fig. 6). At 273 K, RICE-5 exhibited the highest adsorption capacity (~50 cm$^3$ g$^{-1}$ at 1 bar) among the COFs reported in this study. RICE-5 and RICE-6 showcased much higher adsorption capacity than their iso-structural counterparts RICE-3 and RICE-4, which might be attributed to their higher porosity as determined by the nitrogen sorption tests[44]. At 298 K, the adsorption capacities of all COFs RICE-3/4/5/6 decreased significantly. Interestingly, RICE-7 displayed similar CO$_2$ adsorption performance at 273 K and 298 K (approximately 24 cm$^3$ g$^{-1}$ at 1 bar). which might be related to narrow pore channels (Supplementary Figs. 60, 61) due to the access and occupancy of CO$_2$ molecules. RICE-7 exhibited comparable CO$_2$ adsorption performance at 298 K as other 3D COFs reported[39], indicative of its great potential as an effective

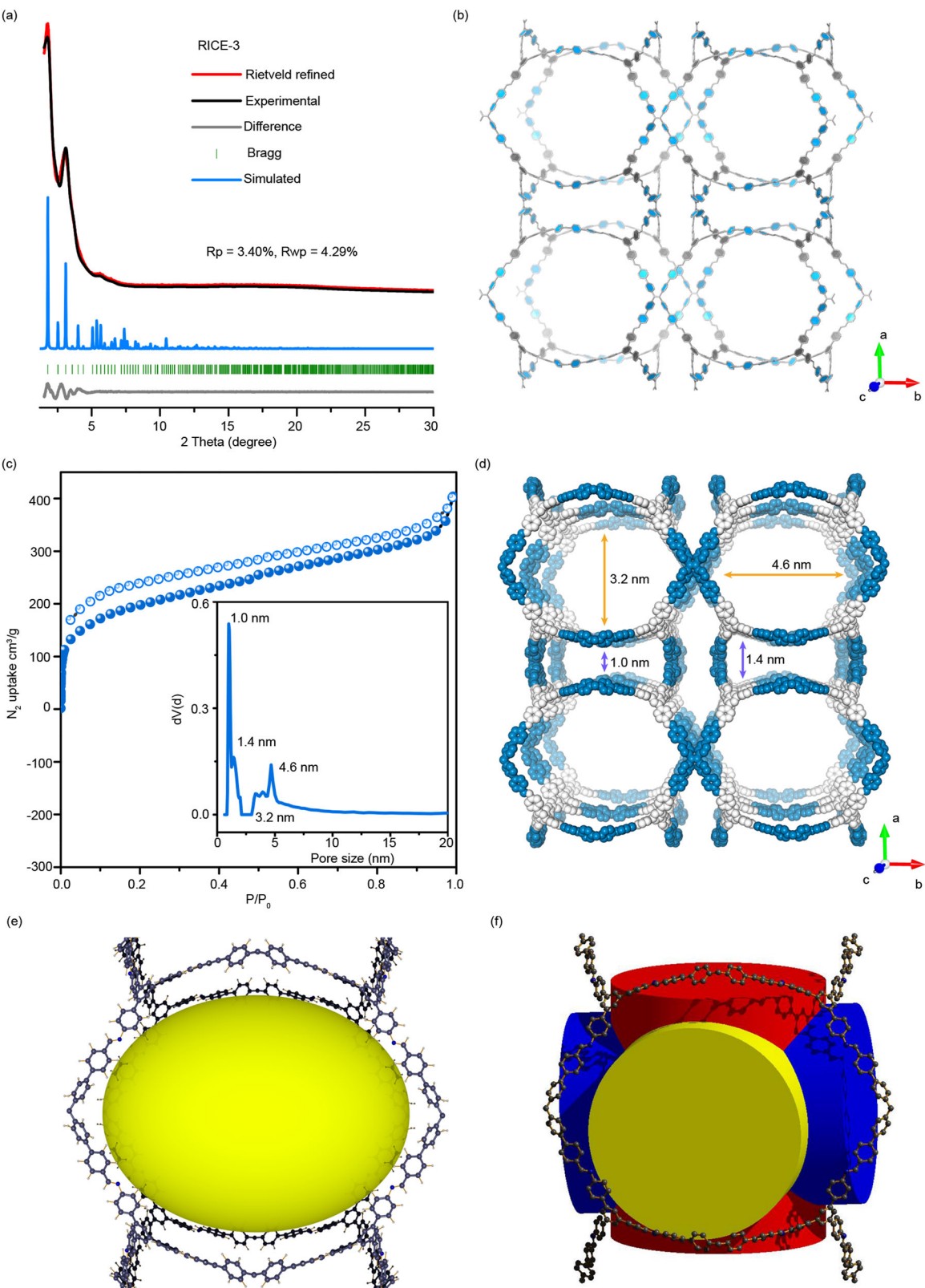

**Fig. 3 | Crystal structures and porosity of RICE-3. a** PXRD patterns and refinement result for RICE-3. **b** Structural representations of RICE-3 in a view of 001 facet. **c** Nitrogen sorption isotherms of RICE-3; inset shows the pore size distributions of RICE-3. **d** Pore visualization of RICE-3. **e** 4.6 nm aperture visualization in RICE-3. **f** Open edge-transitive and highly symmetric feature in pto net.

adsorbent for $CO_2$ removal at room temperature. Density functional theory (DFT) calculation also proves the favorable $CO_2$ adsorption behavior of RICE-7 (see Supplementary Fig. 85 and other details in Supplementary Information).

## Discussion

In conclusion, we successfully designed and synthesized a series of 3D COFs with edge-transitive pto and mhq-z topologies through the reticular synthesis using rectangular-planar (4-c) and trigonal-planar (3-c)

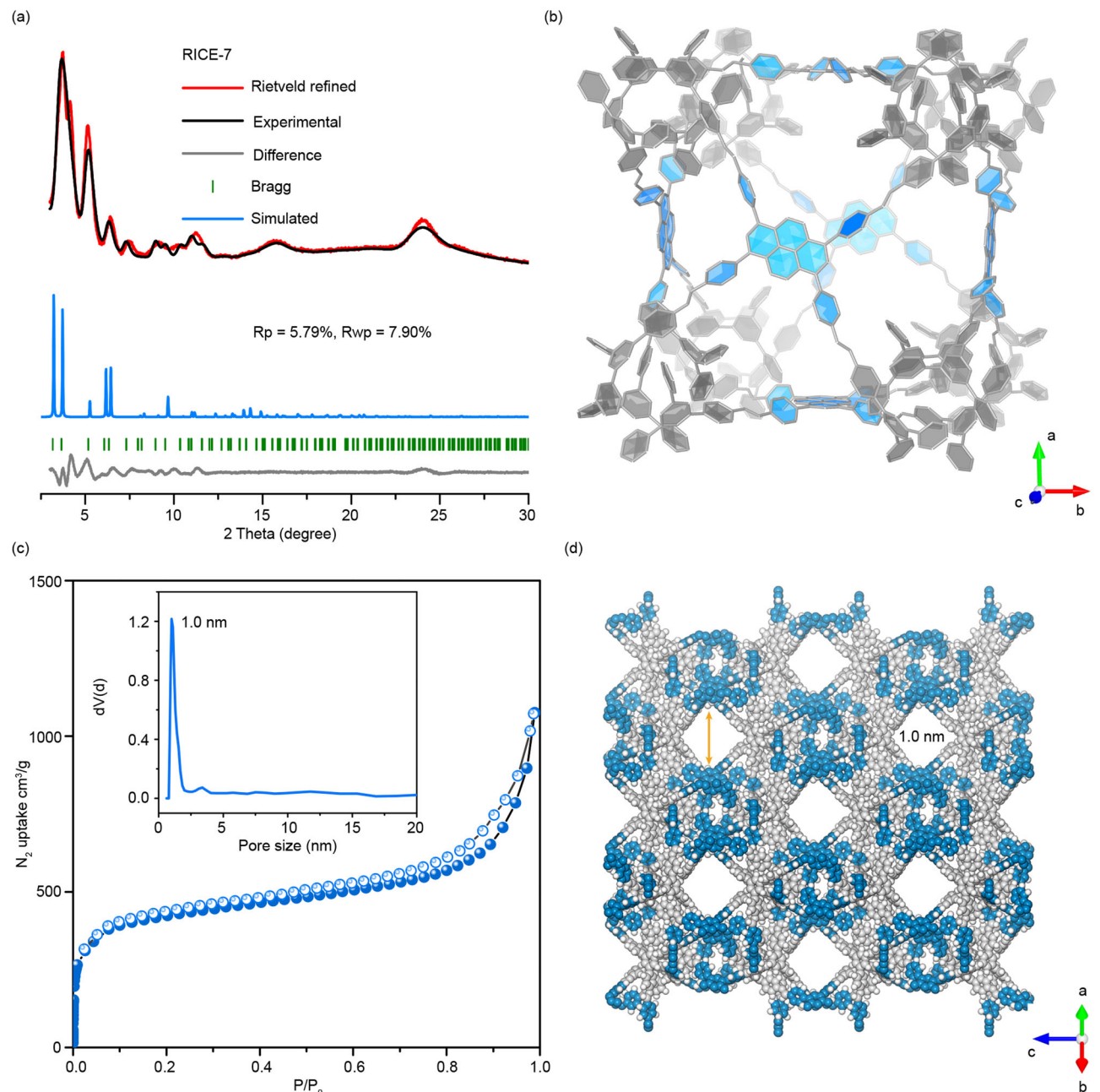

**Fig. 4 | Crystal structures and porosity of RICE-7. a** PXRD patterns and refinement result for RICE-7. **b** Structural representations of RICE-7 in a view of 001 facet. **c** Nitrogen sorption isotherms of RICE-7; inset shows the pore size distributions. **d** Pore visualization of RICE-7.

units. RICE-3/4/5/6 possess a large pore size and low density compared with other framework materials. RICE-7 with mhq-z net displayed three types of totally enclosed truncated polyhedra cages without any open window to form 1D channels, and only formed one channel of 1.0 nm size with the packing of those cages by sharing faces. These 3D COFs demonstrated good $CO_2$ adsorption performance, promising for greenhouse gas capture. This work extends the topological possibilities of 3D COFs by the reticular design of planar monomers and provides more approaches for constructing 3D COFs. This work also advances the design of COF structures for gas capture and other applications.

## Methods
### RICE-3
1,3,5-Tris(4-aminophenyl)benzene (TAPB) (14.06 mg, 0.04 mmol) and 4′,4‴,4‴″,4‴″′-(ethene-1,1,2,2-tetrayl)tetrakis(([1,1′-biphenyl]

−4-carbaldehyde)) (ETTBC) (22.46 mg, 0.03 mmol) were weighed and dissolved in a mixture of 1.6 mL dioxane and 0.4 mL mesitylene in a Pyrex tube directly without degassing. Afterwards, 0.2 mL 6 M acetic acid was added and the solution was sonicated for 10 min. The tube was further sealed, placed in oven and heated under 120 °C for 3 days. All of the products were separated and washed thoroughly using THF and ethanol. The wet powder samples were sealed in a tea bag and dried using the Leica EM CPD300 Critical Point Dryer.

### RICE-4
5′-(4-Aminophenyl)-2′,4′,6′-trimethyl-[1,1′:3′,1″-terphenyl]-4,4″-diamine (ATTA) (7.87 mg, 0.02 mmol) and 4′,4‴,4‴″,4‴″′-(ethene-1,1,2,2-tetrayl)tetrakis(([1,1′-biphenyl]-4-carbaldehyde)) (ETTBC) (11.23 mg, 0.015 mmol) were weighed and dissolved in a mixture of 0.5 mL o-dichlorobenzene (o-DCB) and 0.5 mL 1-butanol (1-BuOH)

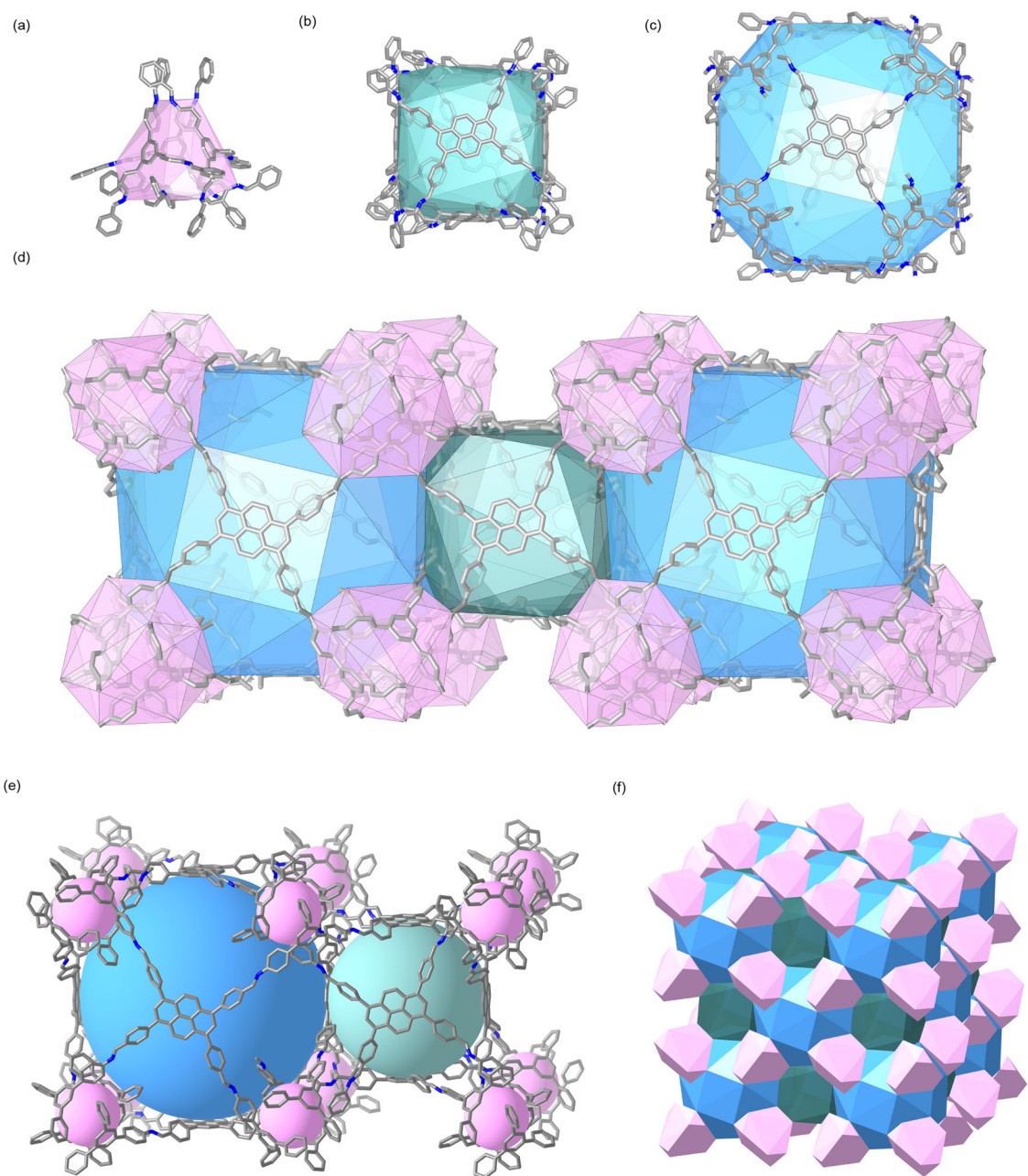

**Fig. 5 | Three types of polyhedra cages in RICE-7. a** truncated tetrahedron **T**$^{cage}$ (pink), **b** small cube **B**$^{cage}$ (green), and **c** cube **C**$^{cage}$ (blue). **d** Representative section of the crystal structure of RICE-7 showing the polyhedra cages. **e** Totally enclosed cages in RICE-7. **f** Representation of a **mhq-z** net, based on a face-sharing truncated polyhedron.

in a Pyrex tube directly without degassing. Afterwards, 0.2 mL 6 M acetic acid was added and the solution was sonicated for 10 min. The tube was further sealed, placed in oven and heated under 120 °C for 7 days. All the products were separated and washed thoroughly using THF and ethanol. The wet powder samples were sealed in a tea bag and dried using the Leica EM CPD300 Critical Point Dryer.

### RICE-5
4′,4‴,4‴″,4‴‴‴-(Ethene-1,1,2,2-tetrayl)tetrakis(([1,1′-biphenyl]-4-amine)) (ETTBA) (10.45 mg, 0.015 mmol) and 1,3,5-tris(4-formylphenyl)benzene (TFPB) (7.81 mg, 0.02 mmol) were weighed and dissolved in a mixture of 0.5 mL 0-dichlorobenzene (o-DCB) and 0.5 mL 1-butanol (1-BuOH) in a Pyrex tube directly without degassing. Afterwards, 0.2 mL 6 M acetic acid was added and the solution was sonicated for 10 min.

The tube was further sealed, placed in oven and heated under 120 °C for 7 days. All the products were separated and washed thoroughly using THF and ethanol. The wet powder samples were sealed in a tea bag and dried using the Leica EM CPD300 Critical Point Dryer.

### RICE-6
4′,4‴,4‴″,4‴‴‴-(Ethene-1,1,2,2-tetrayl)tetrakis(([1,1′-biphenyl]-4-amine)) (ETTBA) (10.45 mg, 0.015 mmol) and 5′-(4-formylphenyl)-2′,4′,6′-trimethyl-[1,1′:3′,1″-terphenyl]-4,4″-dicarbaldehyde (FTTD) (8.65 mg, 0.02 mmol) were weighed and dissolved in a mixture of 0.5 mL o-dichlorobenzene (o-DCB) and 0.5 mL 1-butanol (1-BuOH) in a Pyrex tube directly without degassing. Afterwards, 0.2 mL 6 M acetic acid was added and the solution was sonicated for 10 min. The tube was further sealed, placed in oven and heated under 120 °C for 7 days. All of the products were separated and washed thoroughly using THF and

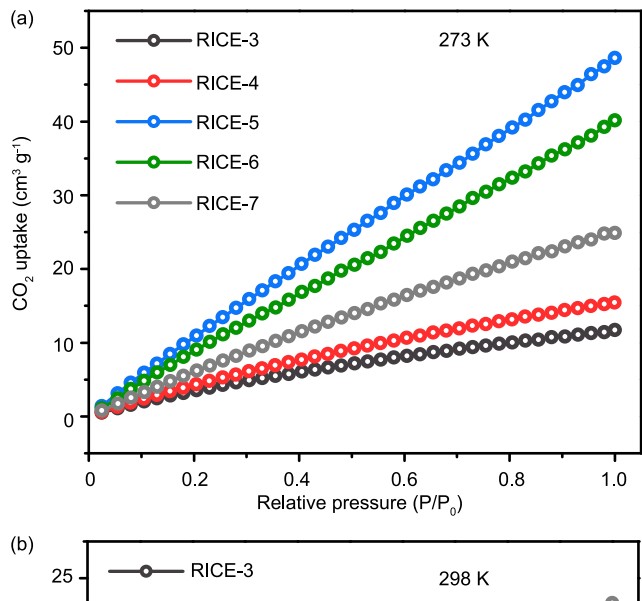

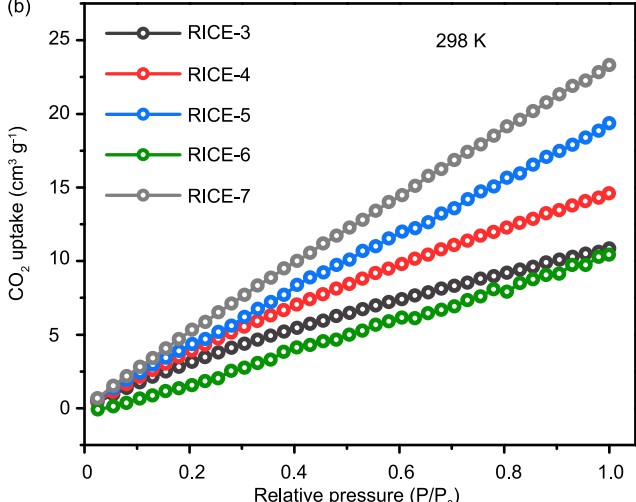

**Fig. 6 | $CO_2$ sorption for RICE-COFs.** $CO_2$ sorption measured at **a** 273 K and **b** 298 K.

ethanol. The wet powder samples were sealed in a tea bag and dried using the Leica EM CPD300 Critical Point Dryer.

## RICE-7

1,3,6,8-tetrakis(4-aminophenyl)pyrene (Py) (17.0 mg, 0.03 mmol) and 1,3,5-tri(4-formylphenyl)benzene (TFPB) (15.6 mg, 0.04 mmol) were weighed and reacted in a mixture of dimethylacetamide/mesitylene/ 6 M acetic acid (5:5:2) in a Pyrex tube. 4 eq. of p-toluidine was added as a modulator which was the key to ensure high crystallinity for the final sample. The tube was further sealed, placed in the oven, and heated under 120 °C for 7 days. The products were separated and washed thoroughly using THF and ethanol.

## Data availability

The data that support the findings of this study are available from the corresponding authors X.Wu and R.V. upon request. Simulated structures for RICE-3/7 have been deposited at the Cambridge Crystallographic Data Centre (CCDC#2259199; 2259205).

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

## Acknowledgements

The authors acknowledge financial support from the Army Research Laboratory (W911NF-18-2-0062) and the Welch Foundation for Chemical Research (C-2124). The authors also acknowledge Shared Equipment Authority at Rice University for access and utilization of characterization instrumentation and the use of Electron Microscopy Center (EMC) at Rice University. X.Wu thanks the National Natural Science Foundation of China (22105202), Natural Science Foundation of Jiangsu Province (BK20200476) and the China Postdoctoral Science Foundation (2021M693178, 2022T150650). The authors acknowledge the Texas Advanced Computing Center (TACC) at The University of Texas at Austin for providing HPC resources that have contributed to the research results reported within this paper.

## Author contributions

D.Z., Y.Z., and X.Wu. conceived and designed the project. D.Z., Y.Z., Q.Y., and X.Wu. synthesized the materials. X.Wu. and H.W. analyzed the PXRD and performed the crystal structure modeling and PXRD Rietveld refinement. X.W. and Y.P. collected and analyzed the transmission electron microscope images. Y.C., C.L. and T.P.S. carried out the theoretical simulations. D.Z., Q.Y., L.A., and G.G. collected and analyzed the nuclear magnetic resonance spectroscopy data. R.V., D.Z., Y.Z. and X.Wu. wrote the manuscript, and all authors discussed and revised it together.

## Competing interests

The authors declare no competing interests.
