## [Peer Review File · Nature Communications]

Three-Dimensional Covalent Organic Frameworks with pto and mhq-z Topologies Based on Tri- and Tetratopic LinkersReviewers' Comments:

Reviewer #1:

Remarks to the Author:

The development of three-dimensional covalent organic frameworks (3D COFs) with novel topology is very interesting. By condensing triphenyl benzene derivatives with tetraphenyl ethylene or pyrene derivatives, several COFs were obtained. However, due to serious problems in COF characterization and structure determination process, I could not support the publication in the journal of Nature Communication.

1. According to FTIR and SSNMR spectra, a large amount of aldehyde groups was found in the frameworks, which is quite similar to the Yaghi's 2D COF paper. The authors should have been more careful to exclude the possible sub-stoichiometric 2D net structure.
2. One of the big issues in this manuscript is the crystal structure. As we know, the crystallinity of 3D COFs is quite important for structure determination. Compared with Yaghi's paper, the PXRD patterns of these COFs are very poor and the peaks are very broad, indicating the severe overlap. So the crystal structure determined by the Le Bail fitting refinement is questionable. The author should optimize the synthetic condition to get a better sample and then solve the real crystal structure.
3. Why is the BET surface areas of RICE3/4/5/6 so low? What is the theoretical value?
4. RICE-4 and RICE-6 are isostructural, only the amine and aldehyde of precursors 4 and 6 are exchanged. Why is the specific surface area and the adsorption capacity of carbon dioxide higher of RICE-6 much larger than that of RICE-4?
5. The CO₂ adsorption isotherms of RICE-7 seems to be unchanged in different temperatures. Why?
6. Please explain the fluctuation at the beginning of TGA curves.

Reviewer #2:

Remarks to the Author:

In this manuscript, the authors developed synthetic methods to design new 3-D COF structures. It is well established that COFs mostly form as 2-D structures, whereas the number of reported 3-D structures is relatively few due to the difficulties of synthesizing building units with appropriate symmetry. In this study, the authors effectively synthesized 5 different 3-D COFs based on pto and mhq-z topologies, which also showed reasonable CO₂ adsorption capabilities. Overall, this is a well written manuscript that reports exciting results that are impactful given the challenges of realizing 3-D COFs. On this basis, it is recommended that this paper be accepted for publication provided the authors are able to address the following comments:

- 1) The authors state that rectangular-planar (4-c) and trigonal-planar (3-c) building blocks are selected to form the new 3-D COFs. It would be helpful if the authors clarify these building units in Scheme 2 with the respect to the organic combinations that can form 4-c or 3-c structures. It would also be helpful to add another scheme (or edit Scheme 1) to highlight the location of these building units in general (i.e. pto and mhq-z structures). The current schemes are hard to follow the concept of 3-D structures formed by those selected organic molecules.
- 2) Please add additional explanations for the crystal structures. For instance, how are the facet numbers assigned in Figures 1 and 2?
- 3) One of the most critical comments pertains to page 9 where the authors mention RICE-3 has mesoporous structures because of lower pressure N₂ uptake ($P/P_0 < 1$). The authors need to add references and provide detailed explanation to justify this statement. As I understood, these features in adsorption/desorption profiles can be also be evidence of micropores. Additionally, the pore size distribution of this sample in Figure 3c indicates the large intensity of 1 nm pores, which is evidence for micropores. The authors need to specify this in Figure 1d. This could be related smaller pores than are specified, which could possibly affect the interpretation of the overall structures.

4) It is unclear in the manuscript what is the motivation for testing CO₂ capture, beyond the general interest in this topic. Could the authors expand on why these 3-D materials may be ideal candidates for this application, and also identify the main adsorption sites for CO₂ in the COFs and their CO₂ adsorption energies.

Reviewer #3:

Remarks to the Author:

Summary: The authors have synthesized and characterized several 3D COFs with novel topologies using a 4+3 monomer strategy. These 3D COFs have very low densities and are capable of good carbon dioxide uptake.

Comments: In the NLDFT pore size analysis of RICE-3, there is a large 1.0 nm pore observed. What is this attributed to? The crystal structure models do not show a pore of this size. The BET surface area for this COF is also much lower than would be expected. Is it possible that there is a large fraction of pore collapsed material?

Does the methyl group on the node monomer affect the stability of the COF? The extra steric hindrance should reduce or restrict rotation around the phenyl-phenyl bonds, but does this affect the structural rigidity in any way? COF stability, to both pore collapse and solvent exposure, was not discussed extensively in this manuscript.

The PXRD peaks for RICE-7 are not very well resolved, though the material has very good surface area and a narrow pore size distribution. It seems as though the material is well behaved for adsorption, but poorly crystalline by PXRD. What could be the reason for this?

The authors collected CO₂ isotherms at both 298K and 273K. What is the heat of adsorption for these COFs and CO₂? This can be a valuable parameter to compare these COFs with others in terms of their ability to separate CO₂ from other gases. One would expect that the large pores would make this a poor material for CO₂ separation, but the large adsorption capacity may suggest otherwise. HOA calculations may add to this discussion.

Conclusions: The synthesis of novel COF structures is both challenging, and fundamentally interesting to the porous materials and polymer communities. After addressing these comments, this referee recommends publication in Nature Communications.

Manuscript titled “**Three-Dimensional Covalent Organic Frameworks with pto and mhq-z Topologies Based on Tri- and Tetratopic Linkers**” (Manuscript ID: NCOMMS-22-31004).

Major changes are as follows: (1) We optimized the synthetic conditions, conducted PXRD and TEM characterizations of COFs, and updated the information to confirm the structures and compositions of RICE-COFs. (2) We provided more crystallographic information and analyses to specify 3D pto and mhq-z topologies for RICE-3/7. (3) We conducted control experiments to investigate the influence of methyl groups on structural rigidity toward both pore collapse and solvent exposure. (4) We performed theoretical simulations to identify the main adsorption sites for CO₂ in COFs and the CO₂ adsorption energies.

REVIEWER COMMENTS

Comments, Reviewer #1

The development of three-dimensional covalent organic frameworks (3D COFs) with novel topology is very interesting. By condensing triphenyl benzene derivatives with tetraphenyl ethylene or pyrene derivatives, several COFs were obtained. However, due to serious problems in COF characterization and structure determination process, I could not support the publication in the journal of Nature Communication.

1. According to FTIR and SSNMR spectra, a large amount of aldehyde groups was found in the frameworks, which is quite similar to the Yaghi's 2D COF paper. The authors should have been more careful to exclude the possible sub-stoichiometric 2D net structure.

Response:

The aldehyde groups in FTIR and ssNMR spectra are attributed to the end aldehyde groups residing on the surface of the COF micro-crystal particles, which is unavoidable due to the non-stoichiometric reactions between aldehyde and amine attaching to the surface of COF particles (*J. Am. Chem. Soc.* **2021**, *143*, 10243). The presence of aldehyde groups in COFs has been reported by many researchers (e.g., *Angew.Chem.Int.Ed.*2021,60,5363–5369).

We exclude the sub-stoichiometric 2D nets for our cases based on two key points. First, as we discuss in the manuscript, the simulated XRD patterns for 2D nets significantly deviate from the experimental profiles (Figures S32 and S45). Second, ratios of monomers for sub-stoichiometric 2D net structures are different from our cases. 2D bex net has a 1:1 ratio for aldehyde and amine monomers, and 2D tth net has a 3:2 monomer ratio. However, for the stoichiometric 3D case in our work, the monomer ratio is calculated to be 3:4 from both digested proton NMR and elemental analyses, so we can rule out the possibility of 2D nets for RICE-COFs.

2. One of the big issues in this manuscript is the crystal structure. As we know, the crystallinity of 3D COFs is quite important for structure determination. Compared with Yaghi's paper, the PXRD patterns of these COFs are very poor and the peaks are very broad, indicating the severe overlap. So the crystal structure determined by the Le Bail fitting refinement is questionable. The author should optimize the synthetic condition to get a better sample and then solve the real crystal structure.

Response:

We tried more synthetic conditions and acquired better PXRD patterns for RICE-COFs. Using *p*-toluidine as a modulator to enhance the reversibility of imine bond formation and crystallization, we could obtain a better PXRD profile for RICE-7 (mqh-z net) which showed more discernible peaks without peak overlap (Figure R1, green). In Figure R1, we can index at least 11 peaks from the new PXRD. a

We have added these data (Figure R1) in ESI (see Figure S14) and updated Figure 2 (i.e., Figure R2) in the manuscript.

Figure R1. Updated PXRD patterns for RICE-7.

Figure R2. Le Bail fitting for Fig. 1 using updated PXRD pattern.

Furthermore, we recollected HRTEM images for RICE-7 using the new sample with higher crystallinity. As shown in Figures R3 and R5, we can now clearly observe several crystallites with periodic channel-like features, and the channel widths of 2.0 nm and 0.5 nm agree with the 1D channel distance in crystal structures (details in Figures R4 and R6).

(This information has been updated in Supporting Information; Figures S28-S31)

Figure R3. HRTEM images for RICE-7 show crystallites with a periodic channel-like (channel width: 2.0 nm) feature.

Figure R4. The periodic channel-like feature (channel width: 2.0 nm) in the crystal structure of RICE-7.

Figure R5. HRTEM images for RICE-7 show a periodic channel-like (channel width: 0.5 nm) feature.

Figure R6. Periodic 0.5 nm channel-like feature in the crystal structure of RICE-7.

Currently, we cannot obtain better PXRD patterns for RICE-3~6 (pto net), and the HRTEM images are not qualified to obtain more useful information. We believe that through a combination of PXRD analyses and crystallographic architectures (Figures 1, S32-S44, S57-S63, and S65-S67) and with the aid of comparisons of pore size distributions between modeled structures and the one derived from nitrogen sorption testes, we can affirm that pto topology is the most credible candidate structure for RICE-3~6.

3. Why is the BET surface areas of RICE3/4/5/6 so low? What is the theoretical value?

Response:

Theoretical surface areas for RICE3/4/5/6 are 8326, 7960, 8382 and 7948 m²/g, respectively (Calculated by Zeo++, *Micropor. Mesopor. Mat.* **2012**, 149, 134). The significant difference between theoretical and experimental values can be attributed to trapped guest molecules, lower crystallinity in partial areas, π/π and the lack of layer interactions in 3D COFs which leads to a higher dynamic compared to 2D COFs. These influencing factors have been reported to significantly impact the accessible surface by N₂ and lead to lower BET surface areas. (*J. Am. Chem. Soc.* **2020**, 142, 14357).

4. RICE-4 and RICE-6 are isostructural, only the amine and aldehyde of precursors 4 and 6 are exchanged. Why is the specific surface area and the adsorption capacity of carbon dioxide higher of RICE-6 much larger than that of RICE-4?

Response:

Isostructural COFs differ in crystallinity, porosity (nitrogen sorption ability; specific surface area), and CO₂ adsorption capacity possibly due to the different reactivities of amine and aldehyde monomers. This phenomenon has been indicated by many prior publications. For example, COF-300 and 3D-IL-COF-1 are isostructural and their surface areas are 1360 m² g⁻¹ and 517 m² g⁻¹, respectively. COF-320 and 3D-IL-COF-2 are isostructural and their surface areas are 2400 m² g⁻¹ and 653 m² g⁻¹, respectively. COF-PTPA and IISERP-COF-4 are also isostructural and their surface areas are 579 m² g⁻¹ and 337 m² g⁻¹, respectively.

So, the higher reactivity of monomers will lead to higher crystallinity for COFs, and thus the higher specific surface area and higher CO₂ adsorption capacity. And it has been widely recognized that higher crystallinity in MOF/COF contributes to a higher surface area and gas sorption behavior (*J. Am. Chem. Soc.* **2018**, *140*, 16124).

Scheme R1. Isostructural COFs. COF-300: *J. Am. Chem. Soc.* 2009, *131*, 13, 4570–4571; 3D-IL-COF-1 and 3D-IL-COF-2: *J. Am. Chem. Soc.* 2018, *140*, 13, 4494–4498; COF-320: *J. Am. Chem. Soc.* 2013, *135*, 44, 16336–16339; COF-PTPA: *Chemistry Select* 2018, *3*, 13442–13455; IISERP-COF-4: *J. Mater. Chem. A*, 2018, *6*, 17307-17311.

5. The CO₂ adsorption isotherms of RICE-7 seems to be unchanged in different temperatures. Why?

Response:

The CO₂ adsorption isotherms for RICE-7 are shown in **Figure R7**. We can observe that the adsorption capacity at 273 K is slightly higher than that at 298 K. Usually, adsorption capacity at lower temperatures is higher than that at higher temperatures.

To note, RICE-7 has a unique structure whose entire framework is built by connecting enclosed truncated polyhedron cages without any open windows to form 1D channels. Packing of those cages by sharing faces (Figures 2/3 and **R8/9**, views of RICE-7 in different vectors) only formed one channel with a diameter of 1.0 nm. During the adsorption process, the 1.0 nm channels will be further narrowed after the access and occupancy of CO₂, so the pathway for

CO₂ diffusion at 273 and 298 K may not show any significant difference, resulting in similar CO₂ adsorption capacities at 273K and 298 K.

Furthermore, we calculated the CO₂ adsorption heat (*Clausius-Clapeyron* equation) for RICE-7 (Figure R10), which is below 5 kJ/mol. This value is pretty low and can be classified as physical adsorption which only happens on the surface of COF particles.

(We have updated this information in the main text and Supporting Information as highlighted)

Figure R7. CO₂ adsorption curves for RICE-7.

Figure R8. Pore visualization from 001/100 facet for RICE-7.

Figure R9. Pore visualization from the 110 facet for RICE-7.

Figure R10. Heat of CO₂ adsorption for RICE-7.

6. Please explain the fluctuation at the beginning of TGA curves.

Response:

The fluctuations at the lower temperature range of TGA curves were caused by the testing procedures and electrostatic charges on the COF surface. COFs are very lightweight (density below 0.1 g cm⁻³) and have electrostatic charges on the surface, so the powders can easily float when nitrogen flows into the testing chamber at the initial stage, resulting in fluctuations at the lower temperature range. We re-tested the samples several times and obtained better curves as shown in Figure R11.

(We have updated TGA curves as shown in Figure S11)

Figure R11. TGA curves for COFs RICE-3/4/5/6/7.

Comments, Reviewer #2

In this manuscript, the authors developed synthetic methods to design new 3-D COF structures. It is well established that COFs mostly form as 2-D structures, whereas the number of reported 3-D structures is relatively few due to the difficulties of synthesizing building units with appropriate symmetry. In this study, the authors effectively synthesized 5 different 3-D COFs based on pto and mhq-z topologies, which also showed reasonable CO₂ adsorption capabilities. Overall, this is a well written manuscript that reports exciting results that are impactful given the challenges of realizing 3-D COFs. On this basis, it is recommended that this paper be accepted for publication provided the authors are able to address the following comments:

1. The authors state that rectangular-planar (4-c) and trigonal-planar (3-c) building blocks are selected to form the new 3-D COFs. It would be helpful if the authors clarify these building units in Scheme 2 with the respect to the organic combinations that can form 4-c or 3-c structures. It would also be helpful to add another scheme (or edit Scheme 1) to highlight the location of these building units in general (i.e. pto and mhq-z structures). The current schemes are hard to follow the concept of 3-D structures formed by those selected organic molecules.

Response:

Per this reviewer's suggestion, **we added 4-c and 3-c building units in Scheme 2 to make the 4-3 combinations clearer as shown in Figure R11. We also revised Scheme 2 accordingly in the manuscript.**

To highlight the location and better demonstrate the crystal structures for pto topologies, we add Figure **R12** (added as revised Figures 2e-2g in the revised manuscript) for RICE-3. Figure **R12** is specifically provided here to show that pto is a highly symmetric edge-transitive and open/porous 3D structure, which shows the same pore structures when viewed from different vectors.

Fig. R11. Synthesis of RICE-3/4/5/6/7

Figure R12. Updated visualization of pto topology for RICE-3.

For RICE-7 with the mqh-z topology, it showcases highly reticular complexity, so it is unlikely to edit Scheme 1 to fully clarify their building units in just one figure that is viewed from one vector. Therefore, we specify its unique structure in Figure 4 in the manuscript to fully demonstrate its building units, pores, cages, shared faces, and packing models.

2. Please add additional explanations for the crystal structures. For instance, how are the facet numbers assigned in Figures 1 and 2?

Response:

We have added the facet numbers (001) and (111) as assigned in Figure 1 and Figure 2 to both figures and captions (as shown in Figure R13 here). In the response to **Comment 1 (Reviewer 2)**, we provide additional Figure R12 to make a clearer visualization for pto topology (RICE-3). **More descriptions have been added in the revised manuscript on Pages 8-9.**

Figure R13. Visualization from facet 001 and facet 111 for RICE-3 and RICE-7, respectively

3. One of the most critical comments pertains to page 9 where the authors mention RICE-3 has mesoporous structures because of lower pressure N_2 uptake ($P/P_0 < 1$). The authors need to add references and provide detailed explanation to justify this statement. As I understood, these features in adsorption/desorption profiles can also be evidence of micropores. Additionally, the pore size distribution of this sample in Figure 3c indicates the large intensity of 1 nm pores, which is evidence for micropores. The authors need to specify this in Figure 1d. This could be related to smaller pores than are specified, which could possibly affect the interpretation of the overall structures.

Response:

Thanks for this insightful suggestion. After careful examination, we confirm that the sorption isotherm of RICE-3 shows both micropores and mesopores. The N_2 adsorption capacity increases rapidly at the lower relative pressure range ($P/P_0 < 0.1$), characteristic of microporous materials. The hysteresis can be observed at $P/P_0 > 0.4$, indicative of the mesoporous pores. These nitrogen sorption isotherm features (see Figure R14) were thoroughly discussed in a prior report (Acc. Mater. Surf. Res, 2018, 3(2): 34-50.). **We have revised relevant statements in the revised manuscript on Page 8 as highlighted.**

As displayed in Figure 1, many micropores areas (1-1.4 nm) exist in RICE-1. We note that previous reports also demonstrated the coexistence of micropores and mesoporous features in 3D COFs (*J. Am. Chem. Soc.* **2020**, *142*, 13334).

Figure R14. Two typical nitrogen sorption isotherms for micro/mesoporous materials and mesoporous materials. Reference: Acc. Mater. Surf. Res, 2018, 3(2): 34-50.

4. It is unclear in the manuscript what is the motivation for testing CO₂ capture, beyond the general interest in this topic. Could the authors expand on why these 3-D materials may be ideal candidates for this application, and also identify the main adsorption sites for CO₂ in the COFs and their CO₂ adsorption energies.

Response:

The massive combustion of fossil fuels is increasing relentlessly accompanied by the excessive release of CO₂, which leads to energy depletion, environmental pollution, and the greenhouse phenomenon, so this prompts us to study whether 3D COF materials can address this challenge by utilizing their unique traits such as excellent functional group tolerance, uniform atomic structures, and high structural tunability. RICE-COFs showcase large void spaces and abundant open channels and may serve as promising materials for gas uptake, so we are interested in investigating their CO₂ uptake performance.

We also applied density functional theory (DFT) to compute the CO₂ adsorption strength on potential adsorption sites within the COF structure. The COF structures in the manuscript have too many atoms and are difficult to be calculated using the DFT method, so we were only able to compute the adsorption behavior of RICE-7. All attempts to optimize structures featuring the chemisorption of the CO₂ molecule to the COF framework relaxed back into linear molecular structure. Thus, we concluded that CO₂ uptake can be attributed to CO₂ physisorption within the COF pores. To assess different physisorption sites, we placed the CO₂ molecule at different hollow sites along the COF pores, as well as in different orientations. A CO₂ molecule placed at the center of the largest pore does not interact strongly with the COF framework given the size of the pore (i.e., the molecule's position falls at least 13 Å from the COF), so we took this structure as the reference for zero interaction energy. When CO₂ is placed closer to benzyl and imine functional groups of RICE-7 as shown in Figure R15, the adsorption energy is more favorable by -0.12 eV. This indicates favorable physisorption as CO₂ is stabilized by its interaction with the functional groups of the COF material via van der Waals

interactions. As we further manipulated the molecular orientation to make the C=O bond in CO₂ in parallel to the C=C bond and C=N bond in the RICE-7 framework, the adsorption energy is as strong as -0.45 eV (-43.4 kJ/mol). This adsorption strength is comparable to CO₂ adsorption in zeolites (J. Phys. Chem. C 2012, 116 (19), 10692–10701) and metal–organic frameworks (MOFs) (J. Phys. Chem. Lett. 2014, 5 (5), 861–865; Chem. Sci. 2014, 5 (12), 4569–4581), which are known CO₂ absorbents. These calculations show that RICE-7 is a suitable material for CO₂ capture applications.

We have added this discussion to the main text and Supporting Information as highlighted in yellow.

Figure R15. Representative CO₂ adsorption configurations in the COF RICE-7 structure at various hollow sites. O, C, N, and H are shown as red, grey, blue, and white spheres, respectively. The computed adsorption energies referenced to panel (a) are listed in the figures.

Comments, Reviewer #3

The authors have synthesized and characterized several 3D COFs with novel topologies using a 4+3 monomer strategy. These 3D COFs have very low densities and are capable of good carbon dioxide uptake.

1. In the NLDFT pore size analysis of RICE-3, there is a large 1.0 nm pore observed. What is this attributed to? The crystal structure models do not show a pore of this size. The BET surface area for this COF is also much lower than would be expected. Is it possible that there is a large fraction of pore collapsed material?

Response:

The crystal structure model for RICE-3 has micropores in the range of 1.0 - 1.4 nm, we have revised Figure 1 to specify pore size at 1.0 nm (Figure R16). Several factors may result in a lower surface area than expected, such as trapped guest molecules, lower crystallinity in partial regions, or/and the lack of layer interactions in 3D COFs which leads to a higher dynamic compared to 2D COFs, which have been discussed in a prior report. (*J. Am. Chem. Soc.* **2020**, *142*, 14357).

Figure R16. Pore visualization for RICE-3

2. Does the methyl group on the node monomer affect the stability of the COF? The extra steric hindrance should reduce or restrict rotation around the phenyl-phenyl bonds, but does this affect the structural rigidity in any way? COF stability, to both pore collapse and solvent exposure, was not discussed extensively in this manuscript.

Response:

The methyl group on the node monomer may affect both pore collapse stability and solvent immersion stability of COFs. Our prior report (Verduzco et al. *Chem. Sci.*, 2022,13, 9655-9667) indicated that pore substituents could significantly affect the pore collapse stability of COFs. In general, large dimensions of substituents increase the pore collapse stability of COFs, but COFs with small-size substituents may also be fragile. Another prior report (Wu et al. *J. Am. Chem. Soc.* 2018, 140, 47, 16124–16133) showed that modulation of steric hindrance generally

enhanced the chemical stability (or solvent immersion stability) of COFs under extreme conditions, but most imine COFs were very stable under ordinary solvent immersion.

To investigate the difference in pore collapse stability and solvent immersion stability between RICE-3 (without methyl group) and RICE-4 (with methyl group), we conducted solvent activation and immersion experiments for RICE-3 and RICE-4. Experimental details were added in the Supporting Information. Results show that both RICE-3 and RICE-4 (Figure R18) have excellent solvent stability, which is similar to most imine COFs. Figure R17 indicates that RICE-3 and RICE-4 are stable upon activation with hexane and ethanol, which have lower surface tension solvents, and they collapse after activation with THF which has high surface tension. This result indicates that RICE-3/4 showcases similar pore collapse stability possibly because the small size of the methyl group does not have a significant impact on the structural rigidity of COFs.

Meanwhile, we also performed the stability experiments of RICE-7 by immersing COF powders in different solvents including DCM, THF, DMF, and 1 M HCl/NaOH for 72 h. They showcased high stability as confirmed by nearly unchanged PXRD patterns and FT-IR spectra before and after tests. (Figure R18).

(We have added this information in Supporting Information)

Figure R17. PXRD patterns for the solvent exposure tests of RICE-3/4.

Figure R18. PXRD and FT-IR results for the stability tests of RICE-3/4/7.

3. The PXRD peaks for RICE-7 are not very well resolved, though the material has very good surface area and a narrow pore size distribution. It seems as though the material is well behaved for adsorption, but poorly crystalline by PXRD. What could be the reason for this?

Response:

The poor crystallinity of RICE-7 may be attributed to the lower crystallinity in partial regions and the lack of layer interactions in 3D COFs which leads to a higher dynamic compared to 2D COFs.

We tried more synthetic conditions and obtained better PXRDs for RICE COFs by using *p*-toluidine as a modulator to enhance the reversibility of imine bond formation and crystallization. In Figure R1, we show a better PXRD for RICE-7 (mhq-z net) with more discernible peaks without peak overlap (**Figure R1, green**). In Figure R1, we can index at least 11 peaks from the new PXRD. These data (Figure R1) have been updated in ESI and Figure 2 (Figure R2) in the manuscript.

Figure R1. Updated PXRD patterns for RICE-7.

Figure R2. Le Bail fitting for Fig. 1 using updated PXRD pattern

We also recollected HRTEM images for RICE-7 using the new sample. As shown in Figures R3 and R5, we can now clearly observe several crystallites with periodic channel-like features, where channel widths of 2.0 nm and 0.5 nm are in good agreement with the 1D channel distance on crystal structures (details in Figures R4/R6).

Figure R3. HR-TEM images for RICE-7 showing crystallites with periodic channel-like feature (channel width=2.0 nm).

Figure R4. Periodic channel-like feature (channel width=2.0 nm) in crystal structure of RICE-7.

Figure R5. HR-TEM images for RICE-7 showing periodic channel-like feature (channel width=0.5 nm)

Figure R6. Periodic channel-like feature (channel width=0.5 nm) in the crystal structure of RICE-7.

4. The authors collected CO₂ isotherms at both 298K and 273K. What is the heat of adsorption for these COFs and CO₂? This can be a valuable parameter to compare these COFs with others in terms of their ability to separate CO₂ from other gases. One would expect that the large pores would make this a poor material for CO₂ separation, but the large adsorption capacity may suggest otherwise. HOA calculations may add to this discussion.

Response:

We calculated HOA for RICE-COFs (Figure **R19**) based on the *Clausius-Clapeyron* equation. As pointed out by this reviewer, HOA for RICE-3/4 are much lower (below 4 kJ/mol) than RICE-5/6 (27.5 kJ/mol for RICE-5 and 40kJ/mol for RICE-6), suggesting that large pores in RICE-3/4 have a negative effect on CO₂ adsorption, but RICE-5/6 exhibit higher CO₂ adsorption capacity due to their higher porosity. RICE-7 showed a low HOA of below 5 kJ/mol, which may be due to its unique structure as we explained in **Comment 5, Reviewer 1**. We speculate that the lower adsorption behaviors of RICE-3/4/7 may be classified as physical adsorption, which only happens on the surface of COF particles. However, the remarkable

differences between these RICE-COFs also indicate that other unknown factors may also contribute to CO₂ uptake beyond pore architectures, porosity, and crystallinity.

(We have updated this information in Figure S77)

Figure R19. HOA of CO₂ for RICE-COFs

Reviewers' Comments:

Reviewer #1:

Remarks to the Author:

The author did make some progress in the revised manuscript. However, I still have a big concern about the crystal structure. As the author mentioned, the theoretical surface areas for RICE3/4/5/6 are really high (8326, 7960, 8382 and 7948 m² /g, respectively), which is much higher than the experimental data. This big difference cannot be explained just by the trapped guest molecules, lower crystallinity in partial areas and so on (For most simulated crystal structure of 3D COFs, people used this explanation, but this may be not true). I think there is something wrong about the crystal structures, otherwise the results should be almost the same. In recent years, some big achievements in 3D COFs structure determination have been made, such as the growth of single crystal of 3D COFs for SCXRD. The authors should try more reaction conditions to get highly crystalline samples and then solve the crystal structure directly.

Reviewer #2:

Remarks to the Author:

The authors have done a good job of addressing concerns raised in the original review. In the revised manuscript, the authors address all four of the comments from my review. I have no additional requests for edits and believe the current version of this manuscript is sufficient for publication. I suggest accepting the manuscript in its current form.

Reviewer #3:

Remarks to the Author:

It appears that the authors have made significant changes to this manuscript that satisfy this referee's comments. Therefore, this manuscript should now be acceptable for publication in Nature Communications.

Responses for the manuscript titled “**Three-Dimensional Covalent Organic Frameworks with *pto* and *mhq-z* Topologies Based on Tri- and Tetratopic Linkers**” (Manuscript ID: NCOMMS-22-31004A).

=====

REVIEWER COMMENTS

Comments, Reviewer #1

The author did make some progress in the revised manuscript. However, I still have a big concern about the crystal structure. As the author mentioned, the theoretical surface areas for RICE3/4/5/6 are really high (8326, 7960, 8382 and 7948 m²/g, respectively), which is much higher than the experimental data. This big difference cannot be explained just by the trapped guest molecules, lower crystallinity in partial areas and so on (For most simulated crystal structure of 3D COFs, people used this explanation, but this may be not true). I think there is something wrong about the crystal structures, otherwise the results should be almost the same. In recent years, some big achievements in 3D COFs structure determination have been made, such as the growth of single crystal of 3D COFs for SCXRD. The authors should try more reaction conditions to get highly crystalline samples and then solve the crystal structure directly.

Response:

We appreciate the comments provided by this referee. First, we would like to point out that the discrepancy between theoretical and measured porosities is expected and consistent with the prior literature. As an example, dyanCOF-330 was recently reported (Nature Comm, 2022, 13:7936) had an extremely low experimentally measured surface area despite very clear PXRD peaks and HRTEM lattices. We list several additional examples in Table R1. The discrepancy between the experiments and theory can be attributed to a combination of trapped guest molecules, poor crystallinity in specific samples, and weak host-guest interactions (~94% porosity for RICE-3; low heat of adsorption). Therefore, the porosity of these 3D COFs cannot be simply calculated through the N₂ sorption test, and the BET surface areas calculated based on traditional N₂ sorption measurements may have a large difference from the theoretical values.

Table R1. Comparison of calculated/measured BET surface areas of COFs

Type	COF	Calculated (m ² g ⁻¹)	Measured (m ² g ⁻¹)	Measured/Calculated
2D	CCOF-1 (Ref 1) ¹	3890	266	6.84 %
3D	3D-PN-1 (Ref 2) ²	6398	514	8.03%
3D	COF-39 (Ref 3) ³	6680	813	12.17 %
3D	COF-38 (Ref 3)	6951	572	8.23 %
3D	dyanCOF (Ref 4) ⁴	3279	Not detected	--
3D	RICE-3	8326	720	8.65 %
3D	RICE-4	7690	654	8.50 %
3D	RICE-5	8382	649	7.74 %
3D	RICE-6	7948	1612	20.3 %

Regarding single crystal structures of COFs and SCXRD, though we agree with this reviewer that SCXRD can enable precise identification of crystal structures, synthesis of single crystal COFs is very challenging. It may not be possible to produce COF single crystals due to COF precipitation. Currently, single crystal COFs are still rarely reported and only restricted to several types. The crystallization problem is a significant challenge in the COF field as described by Yaghi and coworkers (*Science*, 2018, 361, 48). We followed many synthetic conditions and tried nearly all reported methods including addition of modulators, but the crystallinities for RICE-COFs are still not sufficient for SCXRD or 3D ED analyses.

Our methods for crystal structure identification and COF characterization follows the recommended practices in reticular chemistry field (*ACS Cent. Sci.* 2020, 6, 1255), which includes a combination of computational modeling, PXRD measurements, TEM analyses and other spectroscopic methods to resolve COF structures. The data collected and presented provides strong support and direct evidence for the crystal structures described in the manuscript and will be understood and accepted by researchers in the COF and reticular chemistry fields.

For these reasons, we believe we provide sufficient and strong evidence for the 3D COF crystal structure.

(1) Wang, X.; Han, X.; Zhang, J.; Wu, X.; Liu, Y.; Cui, Y. Homochiral 2D Porous Covalent Organic Frameworks for Heterogeneous Asymmetric Catalysis. *J. Am. Chem. Soc.* **2016**, 138 (38), 12332–12335. <https://doi.org/10.1021/jacs.6b07714>.

(2) Wang, K.; Kang, X.; Yuan, C.; Han, X.; Liu, Y.; Cui, Y. Porous 2D and 3D Covalent Organic Frameworks with Dimensionality-Dependent Photocatalytic Activity in Promoting Radical Ring-Opening Polymerization. *Angewandte Chemie International Edition* **2021**, 60 (35), 19466–19476. <https://doi.org/10.1002/anie.202107915>.

(3) Jin, F.; Nguyen, H. L.; Zhong, Z.; Han, X.; Zhu, C.; Pei, X.; Ma, Y.; Yaghi, O. M. Entanglement of Square Nets in Covalent Organic Frameworks. *J. Am. Chem. Soc.* **2022**, 144 (4), 1539–1544. <https://doi.org/10.1021/jacs.1c13468>.

(4) Wei, L.; Sun, T.; Shi, Z.; Xu, Z.; Wen, W.; Jiang, S.; Zhao, Y.; Ma, Y.; Zhang, Y.-B. Guest-Adaptive Molecular Sensing in a Dynamic 3D Covalent Organic Framework. *Nature Communications* **2022**, *13* (1), 7936. <https://doi.org/10.1038/s41467-022-35674-8>.

Comments, Reviewer #2

The authors have done a good job of addressing concerns raised in the original review. In the revised manuscript, the authors address all four of the comments from my review. I have no additional requests for edits and believe the current version of this manuscript is sufficient for publication. I suggest accepting the manuscript in its current form.

Response:

We appreciate the positive feedback provided by this referee.

Comments, Reviewer #3

It appears that the authors have made significant changes to this manuscript that satisfy this referee's comments. Therefore, this manuscript should now be acceptable for publication in *Nature Communications*.

Response:

We appreciate the positive feedback provided by this referee.

Reviewers' Comments:

Reviewer #1:

Remarks to the Author:

As I pointed out in the previous report, I still have big concerns about the crystal structures of these COFs. Here are my comments:

1. For dyna330 (Nat. Commun. 2022, 13, 7036), it is porous and has a pore volume of 0.73 cm³/g calculated from the CO₂ adsorption isotherm. The BET surface cannot be calculated from the adsorption curve due to the dynamic nature of this unique COF. For soft porous crystals, their calculation are different.
2. The author mentioned Yaghi (ACS Cent. Sci. 2020, 6, 1255) provided the standard method to characterize the structure of 3D COFs. However, as he mentioned in the paper, "Pawley or Le-Bail methods do not consider atomic coordinates, and there is no structural information in these refinements. similarly good refinement values can be reached for different unit cells. Consequently, applying a Pawley or Le-Bail refinement is not informative.....we discourage reporting Pawley or Le-Bail fittings..... The validation of a crystal structure determined by means of powder diffraction is made with the completion of a Rietveld refinement." In the paper, the authors used the Le-Bail fittings, not Rietveld refinement.
3. It is indeed very difficult to get the single crystal of 3D COFs. However, I think it is possible to obtain high quality sample for 3D electron diffraction. The authors should try more reaction conditions.

Responses for the manuscript titled “**Three-Dimensional Covalent Organic Frameworks with pto and mhq-z Topologies Based on Tri- and Tetratopic Linkers**” (Manuscript ID: NCOMMS-22-31004B).

=====

REVIEWER COMMENTS

Comments, Reviewer #1

1. For dyna330 (Nat. Commun. 2022, 13, 7036), it is porous and has a pore volume of 0.73 cm³/g caculated from the CO₂ adsorption isotherm. The BET surface cannot be calculated from the adsorption curve due to the dynamic nature of this unique COF. For soft porous crystals, their calculations are different.

Response:

We agree with the reviewer on this issue, which has been explained in detail in our last revision. Unfortunately, to the best of our knowledge, no well-established methods have been proposed for such calculations. Most researchers only provide surface areas for reference based on traditional BET method, which is a common problem in the porous materials field.

2. The author mentioned Yaghi (ACS Cent. Sci. 2020, 6, 1255) provided the standard method to characterize the structure of 3D COFs. However, as he mentioned in the paper, “Pawley or Le-Bail methods do not consider atomic coordinates, and there is no structural information in these refinements. similarly good refinement values can be reached for different unit cells. Consequently, applying a Pawley or Le-Bail refinement is not informative.....we discourage reporting Pawley or Le-Bail fittings..... The validation of a crystal structure determined by means of powder diffraction is made with the completion of a Rietveld refinement.” In the paper, the authors used the Le-Bail fittings, not Rietveld refinement.

Response:

We agree with the reviewer that Pawley or Le-Bail fitting is not informative enough to fully reveal the atomic structures of RICE-3/7, and this is why we employed a combination of XRD, BET, NMR for digested COFs, and high-resolution TEM (XRD/BET for all COFs; TEM only for RICE-7). Many COF papers also use these fitting methods combined with other characterizations. Here, we list some examples. Omar Yaghi used Pawley refinement and Le-Bail refinement in several recent publications (Pawley: J. Am. Chem. Soc. 2020, 142, 34, 14450–14454; J. Am. Chem. Soc. 2019, 141, 11420–11424; J. Am. Chem. Soc. 2019, 141, 6848–6852; Angew 2020, 59 (5),

2023–2027. Le-Bail: J. Am. Chem. Soc., 2020, 142, 2218–2221;). Donglin Jiang used Pawley refinement in several recent publications (Chem 2021, 7, 3309–3324; J. Am. Chem. Soc. 2021, 143, 8970–8975). Andrew I. Cooper also used Pawley refinement in several recent publications (Nature 2022, 604 (7904), 72–79; J. Am. Chem. Soc. 2021, 143, 15011–15016).

In addition, while we agree that Rietveld refinement is more informative, performing a meaningful Rietveld refinement requires a sufficient number of well-defined diffraction peaks, which is not always feasible for many COFs. Per this suggestion, we carefully performed the Rietveld refinements for RICE-3/7. As shown in Fig R1, the Rietveld refinements confirm the validity of our proposed structures, as proved by the low residual values (low Rp and Rwp) and acceptable profile differences (Figure R1). We have included Rietveld refinements in the revised ESI as Figure S57.

Figure R1. Rietveld refinements for RICE-3/7.

3. It is indeed very difficult to get the single crystal of 3D COFs. However, I think it is possible to obtain high quality sample for 3D electron diffraction. The authors should try more reaction conditions directly.

Response:

We appreciate the reviewer for this suggestion. By following this suggestion, we tried many synthetic conditions (nearly all reported methods) including addition of modulators, but the crystallinities for RICE-COFs are still not sufficient for SCXRD or 3D ED analyses. We found that the single crystal structures of molecular model compounds for RICE-6/7 (TMTP and PBAD, Figures R2/R3) have been reported and available in CCDC database (CCDC identifier: 2027484 for TMTP and 1503281 for PBAD). By utilizing these crystal structures, we were able to further analyze their conformations. The dihedral angles between units of TMTP and PBAD were found to be 75–90°, which were nearly identical to that of our proposed pto and mhq-z topologies for

RICE-6/7 (Figures R2/R3). We believe that these findings effectively assist the rational interpretation of the conformational design/dihedral angles in our simulated pto and mhq-z structures. **We have included this information in the Supporting Information (Figures S58/S59).**

(a)

(b)

Figure R2. (a) Single-crystal X-ray structure of the model compound for RICE-6 (TMTP) with dihedral angles of 74–90°. (b) Dihedral angles of building units in simulated pto topology of RICE-6.

(a)

(b)

Figure R3. (a) Single-crystal X-ray structure of the model compound for RICE-7 (PBAD) with dihedral angles of 75–85°. (b) Dihedral angles of building units in simulated mhq-z topology of RICE-7.

3. Additional revisions:

During the revision process of this manuscript, two examples of 3D COFs with pore sizes of 4.6nm and 4.7nm were reported, so we further cite these two publications as references 40 and 41 to update the newest information, which is highlighted in the manuscript.

(40) Zhao, Y.; Das, S.; Sekine, T.; Mabuchi, H.; Irie, T.; Sakai, J.; Wen, D.; Zhu, W.; Ben, T.; Negishi, Y. Record Ultralarge-Pores, Low Density Three-Dimensional Covalent Organic Framework for Controlled Drug Delivery. *Angewandte Chemie* **2023**, *135* (13), e202300172. <https://doi.org/10.1002/ange.202300172>.

(41) Ding, J.; Guan, X.; Lv, J.; Chen, X.; Zhang, Y.; Li, H.; Zhang, D.; Qiu, S.; Jiang, H.-L.; Fang, Q. Three-Dimensional Covalent Organic Frameworks with Ultra-Large Pores for Highly Efficient Photocatalysis. *J. Am. Chem. Soc.* **2023**, *145* (5), 3248–3254. <https://doi.org/10.1021/jacs.2c13817>.

Reviewers' Comments:

Reviewer #1:

Remarks to the Author:

In the revised version, the author did Rietveld refinements to confirm the validity of their structures. So add this result in the maintext, not left in ESI.

Responses for the manuscript titled “**Three-Dimensional Covalent Organic Frameworks with pto and mhq-z Topologies Based on Tri- and Tetratopic Linkers**” (Manuscript ID: NCOMMS-22-31004B).

=====

REVIEWER COMMENTS

Comments, Reviewer #1

In the revised version, the author did Rietveld refinements to confirm the validity of their structures. So add this result in the maintext, not left in ESI.

Response:

We have done as requested by Reviewer #1. The Rietveld refinement was moved to the main manuscript.